# Transcriptional Regulation of Differentiation and Functions of Effector T Regulatory Cells

**DOI:** 10.3390/cells8080939

**Published:** 2019-08-20

**Authors:** Shin-ichi Koizumi, Hiroki Ishikawa

**Affiliations:** Immune Signal Unit, Okinawa Institute of Science and Technology Graduate University, 1919-1 Tancha, Onna-son, Okinawa 904-0495, Japan

**Keywords:** Treg subsets, effector Treg, transcriptional regulation

## Abstract

Foxp3-expressing regulatory T (Treg) cells can suppress the activity of various types of immune cells and play key roles in the maintenance of self-tolerance and in the regulation of immune responses against pathogens and tumor cells. Treg cells consist of heterogeneous subsets that have distinct phenotypes and functions. Upon antigen stimulation, naïve-like thymus-derived Treg cells, which circulate in secondary lymphoid organs, can differentiate into effector Treg (eTreg) cells and migrate to and control immune homeostasis of peripheral tissues. eTreg cells are heterogeneous in terms of their ability to localize to specific tissues and suppress particular types of immune responses. Differentiation and function of diverse eTreg subsets are regulated by a variety of transcription factors that are activated by antigens and cytokines. In this article, we review the current understanding of the transcriptional regulation of differentiation and function of eTreg cells.

## 1. Introduction

Immune suppressive CD4 T cells expressing the transcription factor Forkhead box protein 3 (Foxp3), known as regulatory T (Treg) cells, play an essential role in maintaining immune tolerance and tissue homeostasis [1]. Foxp3 is indispensable for Treg development, maintenance, and function, and ectopic expression of Foxp3 is sufficient to provide CD4 conventional T cells with immune suppressive functions and Treg phenotypes [1,2,3]. Foxp3 deficiency causes severe autoimmune diseases in both human and mice as shown in IPEX syndrome (immune dysregulation, polyendocrinopathy, enteropathy, X-linked) patients and scurfy mice, respectively [4,5,6].

Treg cells can be subdivided into various unique subsets, and this heterogeneity is essential for Treg-mediated immune homeostasis. The first level of Treg subset classification is based on the developmental pathways. Thymus-derived Treg (tTreg) cells differentiate from CD4/CD8 double-positive or CD4 single-positive thymocytes in the thymus, depending on recognition of tissue-restricted self-antigens expressed in medullary thymic epithelial cells [7,8,9,10]. On the other hand, peripherally derived Treg (pTreg) cells differentiate from naïve CD4 T cells upon antigen stimulation in the presence of TGF-β in the secondary lymphoid tissues [8,11]. Helios and neuropilin-1 (Nrp1) have been widely used as markers to distinguish tTreg (Helios^+^ Nrp1^+^) and pTreg (Helios^-^ Nrp1^-^) cells [12,13]. The roles of tTreg and pTreg cells are distinct, as mice deficient for a *Foxp3* cis regulatory element—conserved non-coding sequence (CNS) 1, in which generation of pTreg cells, but not tTreg cells, is specifically impaired—develop spontaneous inflammation in lung and gastrointestinal tissues [14].

The second level of Treg subset classification is based on Treg’s activation status. tTreg cells can be subdivided into naïve-like central Treg (cTreg) cells (also known as resting Treg cells) and activated effector Treg (eTreg) cells (also known as activated Treg cells or effector memory Treg cells) [15,16,17,18,19,20]. After maturation, tTreg cells egress the thymus as cTreg cells, which are defined by CD62L^high(hi)^CD44^low(lo)^ or CC chemokine receptor (CCR7)^hi^CD44^lo^ phenotypes, circulate in secondary lymphoid organs, depending on the functions of homing receptors CD62L and CCR7 [15,16]. Upon antigen stimulation, cTreg cells differentiate into eTreg cells, which are defined by CD62L^lo^CD44^hi^ or CCR7^lo^CD44^hi^ phenotypes [15,16,21] (Figure 1). eTreg cells express higher levels of Treg effector molecules, such as cytotoxic T cell antigen 4 (CTLA4) and inducible T cell costimulator (ICOS) compared to cTreg cells, which likely contribute to enhanced suppressive activity of eTreg cells [15,16,17,21,22,23,24]. 

The majority of eTreg cells migrate to and accumulate in non-lymphoid peripheral tissues and inflamed sites, probably due to a decrease in CD62L and CCR7 expression and a concomitant increase in the expression of various chemokine receptors (e.g., CCR4, CCR6, and CCR10) and adhesion molecules (e.g., KLRG1, CD103) [16,17,22,23]. eTreg cells can reside in non-lymphoid peripheral tissues, as tissue Treg cells, and play a role not only in the maintenance of immune homeostasis but also in tissue repair and regeneration [25,26,27,28,29,30,31]. For example, visceral adipose tissue (VAT)-Treg cells accumulate in and suppress the inflammation of adipose tissue, thereby regulating insulin resistance [32,33,34]. In the colon, Treg subsets expressing RAR-related orphan receptor (ROR)γt and GATA-binding protein 3 (GATA3) are involved in inhibiting inflammation and tissue repair, respectively [35,36,37]. Thus, each tissue-specific Treg subset exhibits distinct functions and phenotypes, likely due to tissue-specific environmental cues (Figure 2).

A subset of eTreg cells expressing CXC chemokine receptor 5 (CXCR5), known as follicular regulatory T (Tfr) cells, accumulate in the germinal center (GC) of lymphoid organs and suppress GC reactions that are required for high affinity antibody production of B cells [38,39,40]. In contrast, some CXCR5**^+^** Tfr cells with naïve-like phenotypes are present in the blood [41]. Unlike other Treg subsets, GC Tfr cells do not express the α chain of the interleukin (IL)-2 receptor, CD25 [42]. The TCR repertoire of Tfr cells resembles that of Treg cells rather than Tfh cells, consistent with the notion that most Tfr cells do not differentiate from naïve CD4 T cells, unlike Tfh cells [43].

cTreg and eTreg cell maintenance relies on IL-2 and ICOS signals, respectively, in secondary lymphoid tissues [15]; however, some tissue-specific Treg cells, including colonic Treg cells, are maintained in an IL-2-dependent manner [48]. Although most eTreg cells require TCR signals for their maintenance, a small subset of antigen-experienced Treg cells known as memory Treg cells can survive even in the absence of cognate antigens in an IL-7-dependent manner [49,50]. 

Recent studies have revealed substantial heterogeneity in gene expression profiles within each Treg subset, suggesting a considerable functional diversity of Treg cells. In fact, single-cell RNA-sequencing (RNA-seq) analysis of Treg cells has demonstrated the complexity of cellular states in both cTreg and eTreg cells and that eTreg cell diversity is affected by TCR signal strength [51]. In addition, comparative single cell RNA-seq analysis of Treg cells in lymphoid and non-lymphoid (skin and colon) tissues suggested that Treg heterogeneity is associated with the progressive trajectory of Treg states from the lymph nodes to non-lymphoid tissues and the adaptation of Treg cells to each peripheral tissue [52]. Furthermore, mounting evidence indicates that differentiation and functional states of distinct Treg subsets are regulated by various transcription factors (Figure 1 and Figure 2, Table 1). In this review, we focus on the transcription factors that play crucial roles in eTreg differentiation and functions.

## 2. Transcription Factors in a Core eTreg Transcriptional Program

Although eTreg cells, particularly tissue Treg cells, heterogeneously express gene subsets including Treg effector molecules and chemokine receptors (Figure 2), it has been suggested that the differentiation of most eTreg cells depends on a common transcriptional program. In this section, we outline the transcription factors that play a role in the core eTreg transcriptional program. 

### 2.1. IRF4

A member of the interferon regulatory factor (IRF) family of transcription factors, IRF4, regulates differentiation of a variety of immune cells [81]. IRF4 is expressed in eTreg and pTreg cells, but not in cTreg cells, depending on TCR signal [16,21,23]. IRF4 is essential for eTreg generation, as IRF4-deficient Treg cells fail to acquire eTreg phenotypes, including decreased CD62L expression and increased ICOS expression [53,58]. Treg-specific IRF4-deficient mice develop a lethal autoimmune disease characterized by loss of body weight, splenomegaly, lymphadenopathy, and inflammation in lung, stomach, and pancreas [48]. Notably, T helper 2 (Th2)-dependent immune responses are specifically activated in Treg-specific IRF4-deficient mice, indicating an essential role for eTreg cells in suppression of Th2-dependent immunity [53]. Furthermore, IRF4 is required for eTreg generation not only under Th2 inflammatory conditions but also under Th1 inflammatory conditions, suggesting a generalized role for IRF4 in eTreg differentiation [58]. Loss of IRF4 in Treg cells results in defective expression of the majority of eTreg-related molecules, including *inducible T cell costimulatory* (*Icos*), *Il10, Il-1 receptor 11* (*Il1rl1*), *c-maf, fibrinogen-like protein 2* (*Fgl2*), *Ccr8*, and *PR domain containing 1* (*Prdm1*), confirming the necessity of IRF4 in eTreg differentiation [53,58]. Importantly, IRF4 can interact physically and functionally with Foxp3 [53,58]. The colocalization of IRF4 and Foxp3 at the *Icos* locus [53] suggests that IRF4 may cooperate with Foxp3 to regulate eTreg differentiation and function. Thus, IRF4 plays a central role in the eTreg transcriptional program. 

### 2.2. BATF and JunB

IRF4 is thought to interact with AP-1 transcription factors, such as basic leucine zipper ATF-like transcription factor (BATF) and JunB [82], which contain an alpha-helical basic region leucine zipper (bZIP) domain, thereby regulating expression of genes containing AP-1/IRF4 composite elements (AICEs) [83]. Indeed, IRF4, BATF, and JunB colocalize at AICE-containing gene loci in several T cell subsets including Th17 and CD8 cytotoxic T cells [84,85,86]. Furthermore, loss of IRF4 and BATF similarly impair differentiation of Th17 cells, CD8 cytotoxic T cells and T follicular helper (Tfh) cells [86,87,88,89,90,91]. 

Analysis of BATF-deficient mice has shown that BATF promotes Treg accumulation in the colon and visceral adipose tissues (VAT) [54,92]. Furthermore, a recent analysis of mice bearing *Foxp3* A384T mutation, which is found in IPEX patients, revealed a critical role for BATF in eTreg cells [17]. The *Foxp3* A384T mutation in mice causes severe inflammation in the skin, liver, and colon and aberrant activation of Th2 and Th17 immune responses, partly due to reduced expression of BATF. Both *Foxp3* A384T mutant and BATF-deficient Treg cells fail to differentiate into eTreg cells and to accumulate in peripheral tissues [17].

We have recently reported that JunB regulates an IRF4-dependent eTreg transcriptional program [23]. Like BATF and IRF4, JunB is highly expressed in CD62L^–^ICOS^+^ eTreg cells in vivo and can be induced by stimulation with anti-CD3 and anti-CD28 antibodies in the presence of IL-2 in vitro [23]. Treg-specific JunB-deficient mice exhibit severe inflammation in the colon and lung, aberrant activation of Th1, Th2, and Th17 cells, and enhanced humoral immune responses [23]. Unlike IRF4, JunB is not essential for eTreg generation, but JunB is required for homeostasis, colonic accumulation, and TCR-dependent suppressive functions of eTreg cells [23]. JunB regulates expression of a subset of eTreg-related genes, such as *Icos, Klrg1, T cell immunoreceptor with Ig and ITIM domains* (*Tigit*), *Ctla4*, and *Gzmb* in Treg cells in BATF-dependent and BATF-independent manners [23]. Mechanistically, JunB promotes DNA-binding of IRF4 at loci of eTreg-related genes containing AICE motifs, on which JunB colocalizes with BATF and IRF4, such as *Icos* and *Ctla4*. Recently, using different JunB-deficient mouse models, other groups also demonstrated that JunB is essential for the expression of eTreg-related molecules [56,57], although the autoimmune phenotypes of their mice are different from those of our mice [56]. 

### 2.3. Blimp1

B lymphocyte induced maturation protein 1, Blimp-1 (encoded by *Prdm1*) is transcriptionally induced by IRF4 and regulates eTreg functions [50]. Blimp-1 is highly expressed in a subset of eTreg cells producing IL-10 and promotes expression of a subset of IRF4 target genes, including *Il10, Klrg1, Epstein-Barr virus induced gene 3* (*Ebi3*), *Ccr6*, and *B cell leukemia/lymphoma 2* (*Bcl2*), suggesting an important role for Blimp1 in an IRF4-dependent eTreg transcriptional program [58]. Treg-specific Blimp1-deficient mice are healthy at young ages, but they develop autoimmune inflammation in salivary glands and pancreas with age [93,94]. In an experimental autoimmune encephalomyelitis model, Blimp1 was expressed in Treg cells which accumulate in the inflamed central nervous system and contribute to Treg stability by preventing IL-6-dependent induction of DNA methyltransferase 3 α (Dnmt3a), which inhibits Foxp3 expression by methylating the CNS2 of *Foxp3* [95]. Blimp-1 is also preferentially expressed in and is required for the suppressive activity of RORγt^+^ Treg cells by preventing the production of IL-17 inflammatory cytokine [96]. 

### 2.4. Myb

The proto-oncogene myeloblastosis (*Myb*) has been reported as an important regulator of thymus-derived eTreg cells but not pTreg cells [22]. Myb expression is upregulated in ICOS^+^ eTreg subset in vivo and is induced in Treg cells activated with TCR/CD28 stimulation in the presence of IL-2 in vitro [22]. In *Myb*-deficient mice, expression of eTreg-related molecules, such as ICOS, TIGIT, and KLRG1 is severely diminished in Nrp1^+^ tTreg cells [22]. In contrast, Myb is neither expressed in nor required for differentiation of pTreg cells [22]. Unlike Myb, IRF4 is essential not only for eTreg differentiation but also for pTreg differentiation. Furthermore, although IRF4 is critical for the generation of all eTreg cells, Myb is likely required for differentiation of ICOS^+^ eTreg cells from ICOS^-^ eTreg cells, by regulating expression of genes that are associated with eTreg cell survival and proliferation [22]. As in T cell development and Th2 differentiation [97,98], Myb promotes expression of GATA3, but not T-box expressed in T cells (T-bet), in ICOS^+^ eTreg cells [22], suggesting that Myb may also play a role in deciding the fate of GATA3-expressing eTreg subsets. Thus, Myb functions suggest a stepwise transcriptional regulatory mechanism of eTreg differentiation and a key difference between eTreg and pTreg transcriptional regulatory mechanisms. 

### 2.5. NF-κB

The nuclear factor-κB (NF-κB) family is composed of NF-κB1 (p50), NF-κB2 (p52), RelA (p65), RelB, and c-Rel [99]. Two distinct signaling pathways can turn on NF-κB-dependent transcription programs by the inducing active form of NF-κB (dimerized NF-κB): the canonical NF-κb pathway activates RelA/p65 and c-Rel/p65 heterodimers, while the non-canonical NF-κB pathway activates RelB/p52 heterodimers [99]. TCR signal and a co-stimulatory signal cooperate to activate the canonical NF-κB signaling pathway, which promotes conventional T cell activation [100]. The canonical NF-κB signaling pathway has been shown to be critical for the development of Treg cells. Furthermore, recent studies have shown that canonical NF-κB subunits play important roles in eTreg cells [60,101,102].

c-Rel is not only essential for thymic Treg development but also for eTreg functions [59,61]. Deletion of c-Rel in mature Treg cells causes only mild autoimmune phenotypes in aged mice, but it promotes anti-tumor immunity mediated by CTLs [59,61]. In addition, pharmacological inhibition of c-Rel can improve cancer immunotherapy based on immune checkpoint blockage, suggesting that c-Rel is a potential target for cancer immunotherapy [61]. c-Rel-dependent regulation of expression of eTreg-associated molecules, such as *Integrin α E (Itgae), Tigit, Klrg1, Il1r2, and TNF receptor superfamily member 8 (Tnfrsf8)*, may contribute to eTreg-mediated anti-tumor functions, rather than immune homeostasis [61].

RelA also appears to be important for the differentiation and maintenance of eTreg cells. Treg-specific deletion of RelA induces systemic autoimmune disease [59,60,103]. In a competitive setting under non-inflammatory conditions, RelA-deficiency results in significant reduction in eTreg, Tfr, VAT-Treg, and intestinal RORγt^+^ Treg cells [60]. RelA promotes expression of eTreg-related molecules, such as *Il10, Tigit, Icos, Ebi3, Prdm1*, and *Il1rl1*. RelA can be activated by TNF-α and GITR, which may promote eTreg survival [60]. RelA is not required for IRF4 induction and likely works independently of IRF4 in eTreg cells [60]. 

### 2.6. Id2, Id3, E2A, and HEB

E proteins (E2A, HEB, E2-2), which belong to the basic-helix-loop-helix (bHLH) transcription factor family, bind to and regulate genes containing E-box motifs [104]. On the other hand, Id proteins (Id1-Id4) can form heterodimers with E-proteins and suppress their transcriptional activity [104]. Id2 and Id3 are essential for Treg maintenance, and Treg-specific Id2/Id3 double knockout mice are lethal due to multi-organ autoimmunity and show increased Tfr number [62]. Although the specific roles of Id2 and Id3 in eTreg cells are not clear, expression of these proteins is dynamically regulated during eTreg differentiation [62,63]. Almost all cTreg cells express Id3, but eTreg cells can be subdivided into Id3^+^ and Id3^-^ subsets. Id3^-^ eTreg cells highly express ICOS, KLRG1, TIGIT, and CTLA-4, suggesting that loss of Id3 expression is a signature for mature eTreg cells [63]. On the other hand, eTreg cells, but not cTreg cells, highly express Id2 [63]. TCR signaling decreases expression of Id3 and increases expression of Id2 in Treg cells in vitro. A tight regulation of Id protein expression is likely important for Treg cells, as Id2 overexpression impairs stability and the immune suppressive functions of Treg cells [64]. In contrast, a recent study has revealed that E-proteins suppress differentiation of eTreg cells, as E2A/HEB double knockout Treg cells exhibit increased expression of effector Treg signature molecules, such as IRF4, ICOS, CD103, RORγt, and KLRG1, and enhanced eTreg stability and suppressive function. Thus, interaction of Id and E proteins likely control eTreg differentiation [65].

### 2.7. TCF1 and LEF1

T cell specific transcription factor 1 (TCF1 encoded by *Tcf7*) and lymphoid enhancer binding factor 1 (LEF1) are members of the high-mobility group (HMG) family, which has highly conserved HMG DNA-binding domains [105]. These molecules have been shown to regulate not only T cell development in the thymus [105] but also eTreg differentiation [66,76]. cTreg cells homogenously express high levels of TCF1 and LEF1, while eTreg cells heterogeneously express these molecules [66]. TCF1^−^ LEF1^−^ eTreg cells, but not TCF1^+^ LEF1^+^ eTreg cells, highly express eTreg-related molecules, such as *Cd44, Icos, Tigit, Irf4, Gata3, Prdm1,* and *Tbx21*, suggesting that loss of TCF1 and LEF1 correlates with maturation of eTreg cells [66]. On the other hand, a subset of TCF1^+^LEF1^+^ eTreg cells express Tfr signature genes, *Bcl6* and *Cxcr5* [66]. Indeed, Tfr cells can be differentiated from TCF1^+^ eTreg cells [66]. TCF1 binds directly to the *Bcl6* gene locus and regulates its expression, thereby playing an essential role in Tfr differentiation [66]. Treg-specific TCF1/LEF1 double knockout mice show spontaneous systemic inflammation in thyroid and salivary glands, lung, and small and large intestines [66,76]. 

### 2.8. Foxo1

Foxo1, as well as Foxp3, belongs to the Forkhead box family of transcription factors [90]. Inactivation of Foxo1 is an essential process for cTreg cells to differentiate into eTreg cells. In cTreg cells, Foxo1 is constitutively expressed, localized in the nucleus, and positively regulates expression of cTreg-related genes such as *Sell*, which encodes CD62L, and *Ccr7* [18,68]. In contrast, in eTreg cells, AKT-dependent phosphorylation destabilizes Foxo1 and prevents it from accumulating in the nucleus, resulting in the inhibition of expression of cTreg-related genes [18,68]. Indeed, Treg cells expressing a constitutive active form of (Akt-insensitive) Foxo1 mutant cannot downregulate expression of CD62L and CCR7 in Treg cells and thereby loses the ability to accumulate in peripheral tissues, which leads to CD8 T cell-dependent autoimmunity and enhanced anti-tumor immunity [18].

## 3. Transcription Factors Regulating Differentiation of Distinct Subsets of eTreg Cells

eTreg cells generated in lymphoid tissue adopt distinct fates, including differentiating into various tissue Treg and Tfr cells. The differentiation and function of distinct eTreg subpopulations are regulated by various transcriptional programs. In this section, we focus on the transcription factors that contribute to the diversity of eTreg cells. 

### 3.1. T-Bet, GATA3, RORγt, Bcl6 and STAT5

T-bet, GATA3, RORγt, and B cell lymphoma 6 (Bcl6) define cell lineages of Th1, Th2, Th17, and Tfh cells, respectively [106,107,108]. eTreg cells heterogeneously express these transcription factors, which endow eTreg cells with suppressive functions specifically against T helper cells expressing the same transcription factors (Figure 2). 

T-bet is a member of the T-box family of transcription factors [109]. Analysis of T-bet fate mapping reporter mice has shown that T-bet is stably expressed in 50–60% of colonic Treg cells and 20–40% of Treg cells in the lymphoid organs, lungs, small intestine, and liver [45]. Depletion of T-bet^+^ Treg cells in mice results in aberrant activation of Th1 cells without affecting Th2 and Th17 cells [96]. Conversely, specific depletion of T-bet^-^ Treg cells induces aberrant activation of Th2 and Th17 cells but not Th1 cells [45]. Although T-bet-expressing Treg cells have unique immune suppressive functions, Treg-specific deletion of T-bet in mice is not sufficient to develop spontaneous autoimmune inflammation, suggesting that T-bet^+^ Treg’s suppressive activity is independent of T-bet expression under steady-state conditions [45]. However, T-bet-deficient Treg cells exhibit impaired proliferation under Th1 inflammatory conditions [46]. In Treg cells, T-bet can be induced by IFN-γ-STAT1 signaling and regulate the expression of CX3CR3, which likely promotes migration of Treg cells to specific tissues [45,46]. 

GATA3, which belongs to the GATA family that bind to GATA motifs [110], is highly expressed in skin Treg cells (~80%) [70,71] and colonic Treg cells (~30%) [37,71]. A recent study has shown that Treg-specific GATA3-deficient mice develop severe skin inflammation with aberrant activation of a Type 2 immune responses [70]. Another study has shown that GATA3 and T-bet double knockout mice, but not their single knockout mice, develop multi-organ autoimmunity [47]. Like in Th2 cells, IL-4 and IL-33 can upregulate GATA3 expression, which in turn promotes expression of IL-33 receptor ST2 in Treg cells. GATA3 is required for stable Foxp3 expression in GATA3^+^ Treg cells [37,71,111]. 

RORγt is a member of the nuclear receptor family [112]. RORγt is expressed in colonic Treg cells (~65%) and small intestinal Treg cells (~35%) [72]. The majority of RORγt^+^ Treg cells are Nrp1^−^ Helios1^−^ pTreg cells, which are induced by commensal microbiota [35,36]. Probably depending on different environmental cues, Treg-specific RORγt-deficient mice have shown impaired control of distinct types of immune responses [35,36]. A report has shown that Treg-specific RORγt-deficient mice have increased levels of serum IgE at the steady state, and their Th2 immune responses are accelerated under Th2-inducing conditions [35]. Another report has shown that Treg-specific RORγt-deficient mice have elevated production of Th17 and Th1 cytokines, but not Th2 cytokines, in the colon and are more susceptible to trinitrobenzenesulfonic acid-induced colitis [36]. Furthermore, a recent study has shown that a unique property of RORγt^+^ Treg cells, but not Treg’s RORγt itself, is critical for suppression of Th17-dependent gut chronic inflammation [113]. Colonization of mice by *Helicobacter hepaticus* results in accumulation of colonic RORγt^+^ Treg cells bearing TCRs specific to *H. hepaticus* [113]. In Treg-specific RORγt-deficient mice, although there is a mild increase in colonic *H. hepaticus*-specific Th17 cells, gut inflammation is not induced upon *H. hepaticus* colonization [113]. However, depletion of RORγt^+^ Treg cells by deleting the transcription factor c-Maf (further discussed below), which is critical for generation of this cell population, causes significant increase of *H. hepaticus*-specific Th17 cells and severe gut inflammation [113].

Bcl6 is expressed in and required for differentiation of Tfr cells, which control germinal center reactions [38,39]. In Treg-specific Bcl6-deficient mice, in which Tfr cells are not generated, humoral immune responses during virus infection are significantly enhanced, and humoral autoimmunity is spontaneously induced in aged mice [114]. Bcl6 directly regulates CXCR5 expression in both Tfh and Tfr cells [115], which enables Tfr cells to accumulate in the germinal centers together with Tfh cells and to suppress Tfh-dependent humoral immunity [38,39,40]. Signal transducer and activator transcription 5 (STAT5) can inhibit Bcl6 expression directly or indirectly by inducing Blimp-1 and is a negative regulator of Tfh differentiation [116,117,118]. Like Tfh cells, Tfr differentiation is inhibited by IL-2-dependent STAT5 activation. Indeed, significant Tfr differentiation suppression has been observed during viral infection when large amounts of IL-2 are produced [73]. Furthermore, loss of CD25 significantly decreases both the frequency and numbers of cTreg cells, but it does not affect the eTreg population [15,119], suggesting that the homeostasis of cTreg cells, but not eTreg cells, is dependent on IL-2/STAT5 signal. However, whether STAT5 promotes cTreg survival or inhibits eTreg differentiation remains unclear. 

### 3.2. c-Maf

c-Maf (encoded by *Maf*), which belongs to the AP-1 family of transcription factors, regulates the function and development of various T cells [82,120]. c-Maf is essential for induction and maintenance of RORγt^+^ Treg cells and Tfr cells [75,76,77]. c-Maf is highly expressed in CD62L^-^ eTreg cells and intestinal Treg cells and is induced by CD3, CD28 and IL-2 stimulation in vitro [76,77]. As mentioned above, in mice colonized with *H. hepaticus*, generation of *H. hepaticus*-specific RORγt^+^ Treg cells depends on c-Maf [75]. Furthermore, c-Maf-dependent suppression of colonic Th17 responses and IgA production contribute to the maintenance of a healthy gut microbiota [76]. In Treg cells, c-Maf facilitates expression of IL-10, while inhibiting expression of IL-17 [75]. 

### 3.3. STAT3

c-Maf and RORγt expression in Treg cells, as in CD4 T conventional cells, is regulated by signal transducer and activator transcription 3 (STAT3) [35,75]. IL-6 and IL-23, both of which activate STAT3, promote generation of colonic RORγt^+^ Treg cells [35]. Furthermore, IL-27, IL-6, and IL-21 can induce c-Maf expression in Treg cells in vitro in a STAT3-dependent manner [75]. Similar to c-Maf, Treg-specific ablation of STAT3 results in the development of spontaneous colitis with abnormal increase of colonic Th17 cells [78]. STAT3 is also involved in Tfr differentiation, with Treg-specific STAT3-deficient mice exhibiting a severely reduced Tfr cell number [75,79].Mammalian target of rapamycin (mTOR) signal, which is required for Tfr generation, phosphorylates STAT3, which then binds to and likely promotes the expression of *Tcf1*, which is essential for Tfr differentiation [79].

### 3.4. PPARγ and RORα

In addition to the above-mentioned lineage-defining transcription factors for T helper differentiation and their regulatory transcription factors, eTreg cells heterogeneously express peroxisome-proliferator-activated receptor γ (PPARγ) and retinoic acid receptor-related orphan receptor α (RORα) [33,80], which regulate tissue-specific eTreg functions (Figure 2). 

PPARγ, which is an essential transcription factor for adipocyte differentiation, is required for the differentiation and functions of VAT-Treg cells [33]. VAT-Treg cells have been identified as a unique subset of tissue Treg cells that accumulate in and maintain inflammation of adipose tissues, thereby regulating insulin resistance [32,33]. In Treg-specific PPARγ-deficient mice, VAT-Treg cells are not generated, and the effect of pioglitazone, an insulin-sensitizing drug, is diminished, suggesting that VAT-Treg cells are a therapeutic target of diabetes [33]. VAT-Treg differentiation proceeds in a stepwise manner: PPARγ^lo^ Treg cells, which are induced by unknown mechanisms in the secondary lymphoid organs, migrate into adipose tissues and further differentiate into PPARγ^hi^ VAT-Treg cells, depending on adipose tissue-specific signals as well as TCR, Foxp3, and IL33-ST2 signals [44]. TCR-dependent activation of IRF4 and BATF is required for generation of PPARγ^+^ Treg cells, probably due to their essential roles in ST2^+^ eTreg differentiation [54].

A nuclear receptor RORα is also involved in the regulation of tissue-specific functions of eTreg cells. Skin Treg cells express high RORα levels, with Treg-specific Rorα deletion accelerating the inflammation induced by innate lymphocyte 2 (ILC2) and Th2 cells in an atopic dermatitis mouse model [80]. In skin Treg cells, RORα promotes the expression of death receptor 3 (DR3), which is also expressed in and promotes activity of ILC2. RORα-dependent DR3 expression allows skin Treg cells to compete with ILC2 for the DR3 ligand, thereby suppressing ILC2-dependent immunity [80]. It has also been suggested that the RORα-dependent suppression of IL-4 expression is required for the immune suppressive activity of skin Treg cells. Like skin Treg cells, colonic Treg cells express higher RORα levels than lymphoid tissue Treg cells [52], suggesting that RORα may regulate Treg functions in non-lymphoid tissues other than skin. 

## 4. Conclusions and Perspectives

A central driver of eTreg differentiation is a transcriptional program mediated by IRF4, which is activated by TCR. In concert with BATF, JunB, and probably other AP-1 transcription factors, IRF4 regulates expression of a majority of eTreg-related genes. IRF4-induced Blimp1 promotes expression of a subset of eTreg-related genes, such as IL-10. In addition, Myb and NF-κB are also crucial for the unfolding of the TCR-dependent eTreg transcriptional program. Furthermore, inactivation of cTreg-related transcription factor Foxo1 is also required for eTreg differentiation. Heterogeneous expression of these transcription factors in each eTreg cell, probably due to different strength and/or duration of TCR signal, may contribute to the diversity of eTreg phenotypes. Future studies are needed to reveal how these TCR-dependent transcription factors cooperate with each other in eTreg differentiation. 

eTreg cells migrate to targeted tissues, adapt to the tissue environments, and exert specific suppressive functions, depending on the transcription factors that are essential for differentiation of T helper cell lineages (T-bet, GATA3, RORγt, and Bcl6) and tissue-specific cell types (e.g., PPAR-γ). Expression of these transcription factors in eTreg cells can be induced in specific cytokine environments, but the in vivo mechanisms regulating the expression of these transcription factors are not fully understood. Particularly, how diverse eTreg subsets are generated and maintained at the steady state remains an open question.

Like well-characterized CD8 memory T cells, a subset of antigen-experienced Treg cells can survive for a long time after the removal of their cognate antigens and mount stronger suppressive activity upon re-exposure to the same antigens [20,49,50,121]. Like eTreg cells, memory Treg cells also exhibit decreased expression of the homing receptors CD62L and CCR7, and increased expression of CTLA4 and ICOS [49,50]. This phenotypic similarity suggests a possible association between the transcriptional programs of eTreg and memory Treg cells; however, the transcriptional mechanisms underlying memory Treg differentiation remain largely unexplored. 

In addition to the transcription factors discussed in this review, epigenetic modifications are closely associated with eTreg differentiation [55,122,123,124,125,126,127,128]; however, the molecular links between the Treg transcriptional program and epigenetic regulation remain poorly understood. Further elucidation of the roles of eTreg-related transcription factors in gene expression, chromatin accessibility, and histone modifications at the single cell level will allow us to identify the missing link between transcription factors and epigenetic regulation in eTreg cells. 

It has also been suggested that metabolic control is important in eTreg differentiation and function [79,129,130]. For example, mTOR-dependent cholesterol biosynthesis likely promotes the proliferation and expression of CTLA4 and ICOS in TCR-stimulated Treg cells [129], suggesting that mTOR-dependent metabolic pathways play a role in eTreg energy generation. Interestingly, recent studies have demonstrated that specific metabolic pathways and/or metabolites can regulate the epigenetic status of T cells [113,131,132]. Mitochondrial respiration levels are higher in Treg cells than in other T cell subsets, and the loss of mitochondria respiratory chain complex III results in decreased Treg suppressive activity and increased DNA methylation [133], although the relevance of this in eTreg populations remains unclear. It is important to determine how, and to what extent, cellular metabolic pathways or metabolites regulate the activity of eTreg-related transcription factors and epigenetic regulators. 

Thus, further elucidation of transcriptional regulatory mechanisms and their crosstalk with epigenetics and metabolism in the differentiation of eTreg, tissue Treg, and memory Treg cells may contribute towards the development of drugs that target specific Treg functions. 

## Figures and Tables

**Figure 1 cells-08-00939-f001:**
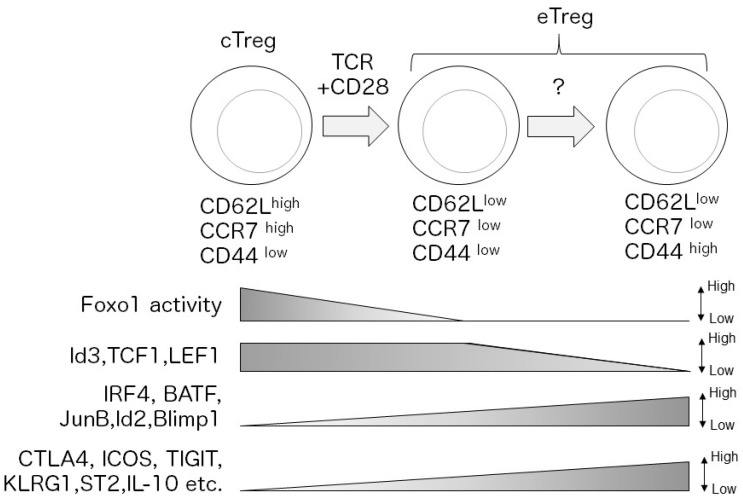
Upon antigen stimulation, central Treg (cTreg) cells (CD62L^hi^ CCR7^hi^ CD44^lo^) differentiate into effector Treg (eTreg) cells (CD62L^lo^ CCR7^lo^ CD44^hi^) depending on TCR and CD28 signaling. After activation, TCR-dependent transcription factors, such as interferon regulatory factor 4 (IRF4), are induced and regulate the eTreg transcriptional program. In contrast, Foxo1 is inactivated by Akt-signaling, which decreases expression of cTreg-related molecules. Loss of Id2, transcription factor 1 (TCF1), and lymphoid enhancer binding factor 1 (LEF1) expression is a signature of mature eTreg cells. Mature eTreg cells highly express immune suppressive molecules, such as cytotoxic T cell antigen 4 (CTLA4) and inducible T cell costimulator (ICOS).

**Figure 2 cells-08-00939-f002:**
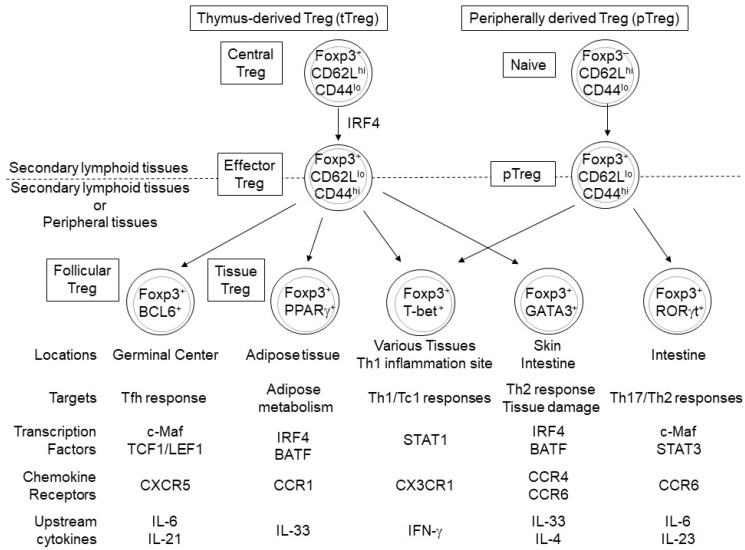
Peroxisome-proliferator-activated receptor γ (PPARγ), B cell lymphoma 6 (BCL6), T-bet, GATA-binding protein 3 (GATA3), and RORγt regulate the differentiation and function of tissue Treg cells derived from eTreg and peripherally derived Treg (pTreg) cells. Treg cells expressing PPARγ, BCL6, or GATA3 mainly differentiate from tTreg cells, while RORγt^+^ Treg cells differentiate from naïve CD4^+^ T cells [34,35,36,43,44] and T-bet^+^ Treg cells likely differentiate from both eTreg and pTreg cells [45,46,47]. These tissue-specific Treg subsets express different functional molecules and play different roles in maintaining tissue homeostasis by suppressing specific immune responses and regulating lipid metabolism and tissue repair.

**Table 1 cells-08-00939-t001:** A summary of eTreg-related transcription factor expression and functions.

Name	Expression	Function	Target	Upstream Regulator	Ref.
IRF4	eTreg	Regulates eTreg differentiation	ICOS, Blimp1, FGL2, ST2, GATA3, CCR8, CTLA4, KLRG1, IL-10, CCR6	TCR+CD28+IL-2	[16,53,54]
BATF	eTreg	Promotes expression of eTreg-related genes	ICOS, GATA3, ST2, KLRG1, CCR4, CCR6	TCR+CD28+IL-2	[17,54,55]
JunB	eTreg	Promotes expression of eTreg-related genes	ICOS, FGL2, CTLA4, TIGIT, KLRG1	TCR+CD28+IL-2	[23,56,57]
Blimp-1	eTreg	Promotes expression of eTreg-related genes	IL-10, KLRG1, Ebi3, CCR6, BCL2	TCR+CD28+IL-2	[58]
Myb	eTreg	Promotes expression of eTreg-related genesin tTreg	ICOS, TIGIT, KLRG1, BCL2, MYC	TCR+CD28+IL-2	[22]
RelA	All Tregs	Promotes expression of eTreg-related genes	TIGIT, ST2, KLRG1, CD103, CD30	TNF-α, GITR	[59,60]
c-Rel	All Tregs	Promotes expression of eTreg-related genes	TIGIT, ST2, ICOS, EBI3,Blimp-1,IL-10	Unknown	[59,61]
Id2	eTreg	Suppresses Tfr differentiation	CXCR5, IL-10, HIF1-α, IKZF3, Myb, IL-10 rα, E2F2	TCR+CD28+IL-2	[62,63,64]
Id3	cTreg, CD44^lo^ eTreg			TCR+CD28+IL-2- Erk,PI3K/mTOR	[62,63]
E2A/ HEB	eTreg	Suppresses eTreg differentiation	ICOS, IRF4, CD103, KLRG1, RORγt	TCR+CD28	[65]
TCF1/ LEF1	cTreg, CD44^lo^ eTreg	Regulates Tfr differentiationSuppresses expression of eTreg-related genes?	CD44, ICOS, TIGIT, IRF4, GATA3, Blimp-1, T-bet, Bcl6, CXCR5	Unknown	[66,67]
Foxo1	All Treg	Maintains expression of cTreg-related genes	CD62L, CCR7, Bim, GzmB	TCR+CD28-Akt	[18,68]
T-bet	eTreg, Tissue Tregs	Regulates migration to Th1-inflammatory sites	CXCR3	IFNγ-STAT1	[45,69]
GATA3	Skin and Intestinal tTreg	Maintains Treg homeostasis Suppresses skin inflammation	ST2, Foxp3	TCR, IL-4, IL-33	[37,70,71]
RORγt	Intestinal pTreg	Regulates migration to intestinal tissue	CCR6	IL-6/IL-23-STAT3, Microbiota	[35,36,72]
BCL6	Follicular Treg	Regulates Tfr differentiation	CXCR5, PD-1	IL-21, IL-6	[73,74]
c-Maf	eTreg	Regulates RORγτ^+^ Treg and Tfr differentiations	RORγt, CXCR5, IL-10	TCR+CD28+IL-2, STAT3, Microbiota, Notch1/2	[75,76,77]
STAT3	All Tregs	Regulates RORγt^+^ Treg and Tfr differentiation	TCF1, RORγt	IL-6,IL-23,IL-21	[35,74,75,78,79]
PPARγ	VAT-Treg	Regulates VAT-Treg differentiation	CCR1, PCYT1A, DGAT1, IL-10	TCR,IL-33, Adipose tissue-derived factor?	[32,34,44,54]
RORα	Skin and Colon Tregs	Promotes expression of eTreg-related genes	CCR6, CD73, DR3	Unknown	[52,80]

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
