# Peer review of "Transcriptional Regulation of Differentiation and Functions of Effector T Regulatory Cells"

_cells, 2019, doi:10.3390/cells8080939_

Round 1

Reviewer 1 Report

Transcriptional regulation of differentiation and functions of effector T regulatory cells

Summary:

This manuscript by Koizumi and Ishikawa is a timely review of the latest findings in transcriptional regulation in eTreg development. This brief review provides a comprehensive discussion of the general functions of a variety of transcriptional regulators in eTreg development and function. If space allows for it, more information could be added to explain the mechanisms by which these factors promote eTreg biology. It would also be appropriate to spend some time pointing out the missing links that exist in eTreg transcriptional regulation and what future studies are required for a more comprehensive understanding of this process. For example, discussion of future studies that relate transcriptional regulation to memory Tregs would be helpful. Overall, this review is a comprehensive examination of the eTreg field and is ready for publication after considering minor revisions (points 4-6,and 8-10 listed below are critical minor revisions while the other points would potentially improve the review but are not critical). 

Minor Points:

1)    Memory Tregs could be discussed here as they certainly fall into a subset of effector Treg cells. The understanding of transcriptional regulation in memory Tregs is not as straightforward but given the tight association between resident tissue Tregs and memory Tregs these are likely related developmentally.

2)    STAT5 could also be discussed regarding its inhibitory function in Tfr development. This could be added in section 3.1 when discussing Bcl-6. Some studies also suggest that STAT5 may be inhibitory for eTreg development as decreasing signals that activate STAT5 cause increased eTreg phenotype.

3)    The authors may consider contrasting the maintenance signaling of effector vs central Treg cells via IL-2, TCR, ICOS and IL7. 

4)    Line 28- These references are incorrect as they describe the role of Foxp3 in Treg development, not the identification of Foxp3, scurfin in IPEX, scurfy disease respectively. Those references should include (Wildin, R. S., F. Ramsdell, J. Peake, F. Faravelli, J. L. Casanova, N. Buist, E. Levy-Lahad, M. Mazzella, O. Goulet, L. Perroni, F. Dagna Bricarelli, G. Byrne, M. McEuen, S. Proll, M. Appleby, and M. E. Brunkow. 2001. X-linked neonatal diabetes mellitus, enteropathy and endocrinopathy syndrome is the human equivalent of mouse scurfy. Nat. Genet.27: 18–20.;  Brunkow, M. E., E. W. Jeffery, K. A. Hjerrild, B. Paeper, L. B. Clark, S. A. Yasayko, J. E. Wilkinson, D. Galas, S. F. Ziegler, and F. Ramsdell. 2001. Disruption of a new forkhead/winged-helix protein, scurfin, results in the fatal lymphoproliferative disorder of the scurfy mouse. Nat. Genet.27: 68–73.; Bennett, C. L., J. Christie, F. Ramsdell, M. E. Brunkow, P. J. Ferguson, L. Whitesell, T. E. Kelly, F. T. Saulsbury, P. F. Chance, and D. Ochs, H. 2001. The immune dysregulation, polyendocrinopathy, enteropathy, X-linked syndrome (IPEX) is caused by mutations of FOXP3. Nat. Genet.27: 20–1.

5)    Line 31- Treg cells likely differentiate from CD4SP as well as perhaps CD4/8 DP cells.

6)    Figure 1- It is unclear what “+α” refers to.

7)    Line 66- It may be worth discussing recent single cell RNA seq analysis of Tregs in reference to the heterogeneity and differentiation trajectories of effector Tregs (David Zemmour, Rapolas Zilionis, Evgeny Kiner, Allon M. Klein, Diane Mathis, and Christophe Benoist. 2018. Single-cell gene expression reveals a landscape of regulatory T cell phenotypes shaped by the TCR. Nature Immunology and Ricardo J. Miragaia, Tomas Gomes, Agnieszka Chomka, Laura Jardine, Angela Riedel, Ahmed N. Hegazy, Natasha Whibley, Andrea Tucci, Xi Chen, Ida Lindeman, Guy Emerton, Thomas Krausgruber, Jacqueline Shields, Muzlifah Haniffa, Fiona Powrie, and Sarah A. Teichmann. 2019. Single-Cell Transcriptomics of Regulatory T Cells Reveals Trajectories of Tissue Adaptation. Immunity)

8)    Figure 2- This figure is somewhat complex and more explanation of the figure in the legend would be helpful. What evidence supports the differential contribution of pTreg or tTreg to different Treg subsets?

9)    Line 82- I don’t recall either of the papers sited using the CD44 vs CD62L paradigm to analyze IRF4-/-Treg cells. These papers analyzed a number of other markers associated with the phenotype of eTreg as well as ability to traffic to/reside in tissues.

10)  Reference 27 explores the cross regulation between Foxp3 and Irf4 and the 2012 Nature Immunology paper (A multiply redundant genetic switch 'locks in' the transcriptional signature of regulatory T cells Fu, W. et al) also explores the contribution of Irf4-Foxp3 cooperation in the Treg signature. These points could be explored here to reference the mechanisms by which Irf4 may contribute to eTreg functionality.

11)  Line 136- Reference 15 also provides data that myb deficiency in Tregs causes decreased GATA3 but not Tbet expression. This is perhaps an interesting point to comment on here given the focus of the review. Further, it has been shown that myb coordinates Th2 differentiation via GATA3 regulation, thus a subset of Tregs may be making use of the same pathway during GATA3+ eTreg development.

12)  Line 200-202: It should be mentioned here that reference 60 show TCF1/LEF1 are critical for Tfr development and that Tfr develop from TCF1+ precursors.

13)  Line 277- This paragraph seems to end abruptly with the notion of STAT3 being required for Tfr generation- is this thought to be through regulation of cMaf or another mechanism? It may be worth discussing the possible roles of STAT3 in eTreg/Tfr generation.

14)  Line 297- Here again, it may be worth expanding on the known and unknown mechanisms regarding RORa in Treg biology.

15)  Line 316- It is worth discussing the current gaps in knowledge concerning the epigenetic modifications that accompany eTreg development and how those relate to the actions of the various transcription factors discussed.

Reviewer 2 Report

The review by Koizumi and Ishikawa is timely given the recent advances in the field, and well balanced. The authors may consider improving the manuscript by addressing the following minor points:

Heterogeneity of tissue Treg cells fat, gut, etc... should be discussed in greater detail, in particular given recent transcriptomic and single cell transcriptomic data addressing tissue adaptation (Miragaia Immunity 2019). The authors should also discuss how metabolism regulates Treg cell fate and function. Usually Wollenberg et al J Immunol 2011 is also cited as a founding reference regarding Tfr cells (together with ref 17 and 18 cited in the document). Some key differences between Tfr and Treg cells should be further discussed namely: Loss of CD25 upon terminal differentiation (Sakaguchi PNAS 2017), IL-2-independence (Botta Nature Immunol 2017) , immature phenotype of Tfr cells in the blood vs tissue (Fonseca Sci Immunol 2017), unique repertoire characteristics (Maceiras Nature Commun 2017).
